# Home-Based Parent–Child Interaction Therapy to Prevent Child Maltreatment: A Randomized Controlled Trial

**DOI:** 10.3390/ijerph18168244

**Published:** 2021-08-04

**Authors:** Mariëlle E. Abrahamse, Vionna M. W. Tsang, Ramón J. L. Lindauer

**Affiliations:** 1Levvel, Academic Center for Child and Adolescent Psychiatry, Meibergdreef 5, 1105 AZ Amsterdam, The Netherlands; v.m.tsang@amsterdamumc.nl (V.M.W.T.); r.lindauer@debascule.com (R.J.L.L.); 2Location Academic Medical Center, Department of Child and Adolescent Psychiatry, Amsterdam UMC, University of Amsterdam, Meibergdreef 9, 1105 AZ Amsterdam, The Netherlands

**Keywords:** PCIT, home-based treatment, attrition, prevention child maltreatment, disruptive behavior problems

## Abstract

High treatment attrition and limited reach of mental health services for at-risk families remains an important problem in order to effectively address the global concern of child maltreatment and child disruptive behavior problems. This study evaluated the effectiveness of a home-based and time-limited adaptation of Parent–Child Interaction Therapy (PCIT). Twenty families with children (70% boys) aged between three and seven years were randomly assigned to an immediate treatment group (IT, *n* = 10) or a waitlist control group (WL, *n* = 10). After receiving treatment and compared to mothers in the WL group, mothers in the IT group reported fewer child behavior problems and more improved parenting skills. Although initial analyses revealed no significant differences, additional analyses showed a significant decrease in the primary outcome of the study, namely child abuse potential, between the baseline and follow-up assessment for the total treated sample. A low treatment attrition rate (15%) was found, indicating higher accessibility of treatment for families. Findings suggest that the brief home-based PCIT is a potentially effective intervention to prevent child maltreatment and disruptive behavior problems in at-risk families. Results also reinforce the importance of addressing the specific needs of these families to increase treatment effectiveness.

## 1. Introduction

Globally, over half of all children—that is one billion children, between two and 17 years of age—have been exposed to some type of physical, sexual, or emotional violence [1]. It is not a surprise that child maltreatment can lead to a range of consequences such as an increased risk for negative development including physical and mental health problems [2]. In the Netherlands, a national prevalence study showed that the rates of child maltreatment for children aged 0 to 17 years were estimated to be 3.4% [3]. Given the prevalence and high impact on child development, early prevention of child maltreatment is therefore of outmost importance. In the past decade, the best efforts of child protection services and increased government attention as well as expenditures in many countries have led to more knowledge about the signs, prevention, and interventions focusing on child maltreatment [4]. Nevertheless, there are some factors impeding the effective prevention and intervention of child maltreatment. First, recognition and help seeking seem to be more difficult for at-risk families [5,6], especially when parents experience a high level of parenting stress [6]. Second, premature treatment attrition remains a significant problem in prevention and intervention such as parent training programs [7,8].

The current study evaluates the effectiveness of an adapted version of Parent–Child Interaction Therapy (PCIT) [9]. Based on theoretical foundations and increasing empirical evidence, PCIT is regarded as one of the most effective treatments in preventing child maltreatment [4,10,11,12]. PCIT is a well-established parent training program originally developed for children aged between two and seven years with disruptive behavior problems, that is widely available across countries and cultures. Since child disruptive behavior problems play an important role in negative parent–child interactions, parenting stress, and parental harsh discipline [13], this behavior is not only a consequence of maltreatment, but also a strong factor in the risk for child maltreatment [4]. Therefore, PCIT has been used at an increasing level in other populations including different ethnic populations [14,15,16] and child welfare populations that are related to child maltreatment. In previous research, PCIT has been found to be effective in reducing further physical abuse in families where physical abuse was present [10,17]. So far, PCIT is not tailored for the specific needs of maltreating parents, and the current discussion in the literature is what this heterogeneous group actually needs to prevent, reduce, or eliminate child maltreatment [18,19]. Therefore, the question is to which degree the improvement of positive parenting skills and the reduction in child disruptive behavior problems is enough to prevent maltreatment in the future. However, the body of empirical evidence is still growing and PCIT has even been found to be an effective treatment to prevent child maltreatment by decreasing child disruptive behavior problems, parenting stress, and by improving parenting skills [4,11,20].

As in many parent training programs, the prevention of attrition remains a big challenge in PCIT and impacts the public health of the families involved. Families who completed PCIT showed significant improvements in the behavior of the child, parenting stress, and parental functioning, while families who dropped out did not achieve the same positive outcomes [21]. Therefore, premature drop-out is considered a serious problem in PCIT, which also affects the effectiveness of the treatment. In previous research, dropout rates ranged from 27 percent to 69 percent [21,22,23,24].

Based on the original treatment protocol [25], PCIT is typically delivered in a playroom at the clinic, with therapists coaching the parents in vivo through a one-way mirror and with a wireless headset. PCIT is performance-based and continues until the treatment goals are met including parents reaching established criteria for both observed skills and their ratings of the child’s behavior within normal limits. Therefore, treatment completion is theoretically equal to treatment success. Delivering treatment in a clinical setting has many benefits such as high environmental control and appropriate equipment. However, it also means that families have to travel to the outpatient clinic. Thereby, in most families referred to community services, other problems are present such as financial and mental health problems, limited motivation, and having too many worries to be able to acquire new skills [26,27,28]. Additionally, simple practical problems such as not having someone to watch the other children during their visit to the clinic, could be a barrier to treatment. Particularly for these families with low resources, it is a challenge to finish a treatment successfully. The literature emphasizes the importance of meeting the needs of families with low resources [29] and a home-based treatment may be most suitable and could be a solution to increase the accessibility of treatments as well as their effectiveness. Treatment delivery in the family’s home comes with several benefits; it eliminates logistic barriers, has a stronger ecological validity since the therapist is observing the parent and child in their natural environment, and also has a faster generalization of the learned skills to other situations [28,29]. In addition, the Centers for Disease Control and Prevention identified home-based programs as the preferred treatment for families at risk for child maltreatment [30].

Several studies have supported the utility of PCIT in the home situation. Home-based PCIT only needs some adjustments to implement and provide the treatment outside a clinical setting, but the core components of PCIT remain intact. Findings of previous studies have shown similar outcomes for home-based and clinic-based groups, suggesting that the added value for home-based PCIT is found in serving a wide range of families who may not have access otherwise. However, the results have been inconclusive about lower rates of treatment attrition [20,21,29,31,32,33,34].

Adapting PCIT for delivery in the home situation is not the only mechanism that can be adjusted to reduce attrition. Previous research has shown positive results from a brief version of PCIT with 12 sessions conducted in families with multiple problems who were at risk for child maltreatment [12]. This protocol was based on the less-is-more principle stating that brief interventions with a moderate number of sessions and a clear end-point are more effective than interventions with a large number of sessions [35]. The clearly defined end-point will encourage families to complete the intervention. In line with the before mentioned research study, a recent meta-analysis has shown that PCIT is effective with and without a specified number of sessions [36].

The current study aimed to bridge the gap between the above-mentioned barriers, particularly the high treatment attrition rate in clinic-based PCIT (40%) [22], and to continue with the ongoing developments on the prevention of child maltreatment in the Netherlands. Hence, the purposes of our study were (1) adapting the original PCIT treatment protocol into a home-based and time-limited version (PCIT-Home) that is suitable for the Dutch mental health service system and (2) evaluating the effectiveness of PCIT-Home for at-risk families. For our effectiveness study, we conducted a randomized controlled trial using a waitlist control condition in a real-world clinical context. The primary outcome was the level of risk for child maltreatment. Secondary outcomes were child disruptive behavior problems, parenting skills, parenting stress, and satisfaction with PCIT-Home. Based on previous research findings on the effectiveness of PCIT for reducing child disruptive behavior, parenting stress, and child abuse potential, and increasing parenting skills, we expected positive effects on these outcomes. Since all these variables are correlates of child maltreatment, we also expected that PCIT-Home would be effective in the prevention of future risk for child maltreatment. Since our adaptation of PCIT is home-based and has a shorter treatment duration with a defined intervention end-point, we additionally expected lower attrition rates than in studies with the clinic-based PCIT.

## 2. Materials and Methods

### 2.1. Participants and Procedure

Twenty families with children aged three to seven years were referred for treatment at a large community mental health center in Amsterdam. Families were primarily referred for disruptive behavior problems and a history of maltreatment of the child. Additionally, for some families, there were concerns about a risk of child maltreatment caused by a disturbed parent–child interaction. For all families, referral took place through usual community channels and most of them were specifically referred for PCIT. Half of the families (45%) were referred by professionals from other community mental health services in the region. These services did not offer PCIT and referring professionals had knowledge in the treatment goals of PCIT and advised families to participate in this intervention. Three families (15%) were referred by their general practitioner or their child’s school, and eight families (40%) were internal referrals from other departments of the mental health center. Recruitment for participation in the evaluation study was between October 2014 and June 2016. Subsequently, the data collection continued until December 2016.

Referred families were selected for inclusion in the study if (1) a risk for child maltreatment or disruptive behavior problems were reason for referral and (2) the referred child was aged between two and seven years. However, serious concerns about the child’s safety in the home situation was an exclusion criterion. In one referred family, the child was removed from home before the start of treatment.

In total, 42 families (see study flowchart, Figure 1) were assessed for eligibility. When referred families met the inclusion criteria, they received information about the purposes and procedures of the study during their intake appointment. They were also informed that the study participation was voluntary and took some additional time investment. Two families were excluded from the study because the parents did not wish to participate. For the other excluded families (*n* = 20), other reasons were present that made them unable to participate in the study (see Figure 1). After the parents provided informed consent (*n* = 20), they were individually randomized to the immediate treatment (IT) group or waitlist control (WL) group using an allocation ratio of 1:1. The randomization list was managed by a researcher with no further involvement in the study. This random assignment resulted in 10 families assigned to the IT group and 10 families assigned to the WL group. With regard to the assessments, the IT group was assessed at three time points and the WL group was assessed at four time points. After randomization, the baseline assessment (T_0_) was scheduled at the family’s home.

All assessments were only completed for female caregivers. To simplify the language in this paper, the term parent will be subsequently used throughout the article. Although including fathers in research studies is valuable, we made the decision to exclusively let mothers complete the assessment, since for all children a female caregiver was involved in treatment. In a previous research study on the effectiveness of PCIT in the Netherlands, fathers were included as participants and they showed similar results as mothers [37]. Based on these similar results and our experiences of including a large group of single mothers and involving fathers asking a large time investment and practical challenges, we decided to only include mothers in our current study.

Two months after the baseline assessment, the families in the WL group were seen for the post-waitlist assessment (T_01_), which served as the pretreatment assessment for this group at the same time. The IT group families were visited after completing the 8-session treatment (T_1_). In sum, the IT group was assessed at three time points and the WL group was assessed at four time points. In the study design, the time between assessments was estimated at two months, therefore the post-waitlist assessment for the WL group took place two months after the baseline assessment. However, in reality the treatment duration was longer, and the average time difference between baseline (T_0_) and post-treatment assessment was 3.4 months (*SD* = 1.7) for the IT group. In addition, the WL group was also visited for the post-treatment assessment (T_1_) after completing treatment. Follow-up assessments (T_2_) for all families were scheduled two months after the post-treatment assessment. Ethical approval for the current study was obtained from the Medical Ethics Committee of the Academic Medical Center of Amsterdam (2014.252).

### 2.2. Parent–Child Interaction Therapy (PCIT)

All families received the home-based adaptation of Parent–Child Interaction Therapy (PCIT) [38], which progresses through two treatment phases. In the first phase, the Child-Directed Interaction (CDI), there was a focus on enhancement of the relationship between the parent and child. In PCIT-Home, families proceeded after four CDI sessions to the second 4-session Parent-Directed Interaction (PDI) phase focusing on behavioral management. Both phases started with a didactic session that was followed by weekly coaching sessions. During CDI, the therapist taught parents to follow their child’s lead during play and coached parents to use specific skills (i.e., praise, reflection, and behavior description). In addition, parents were coached to respond to their child’s appropriate behaviors while ignoring their negative behavior. During PDI, the leading role was switched to the parent, where parents were taught to give clear commands to their child and were coached to provide consistent and safe consequences for non-compliance. Teaching these parenting skills has the largest impact on the child’s behavior when both parents consistently use them. Therefore, including both parents in treatment is the most favorable option. According to the PCIT model, when both parents participate in treatment, they are individually coached while playing with their child, while the other parent is watching. After coaching the first parent, they change positions.

Instead of ending treatment when the parents reached the established mastery-criteria, PCIT-Home ended after eight sessions. For these sessions, the original treatment protocol was used [25]. Since there was no one-way screen available in the home situation, therapists coached parents from a distance using a wireless headset or the therapist took a strategic position behind the parent.

### 2.3. Training and Fidelity

All therapists that provided PCIT-Home were practicing clinicians employed in a community mental health setting. They completed the training process that is currently endorsed by the authorizing body of PCIT, PCIT International, including the initial 40-h workshop and subsequent consultation by a PCIT Global trainer. Consistent with the Dutch and international guidelines for PCIT trainees, all therapists completed higher education with bachelor’s and master’s degrees in mental health fields. For the home-based PCIT, therapists received instructions and additional materials related to issues of providing PCIT in other settings. Additionally, the PCIT Global Trainer provided biweekly consultation to maintain treatment fidelity.

The PCIT protocol recommends that all therapy sessions are videotaped for integrity and supervision purposes. However, due to practical problems (e.g., no permission of parents, lost videotapes, or problems with recoding material), videos from only 50% of the participating families were available for integrity coding. Because the therapists received additional supervision, treatment adherence was also informally assessed. Therefore, only one random treatment session for each available family was coded for integrity. Fidelity checklists from the original treatment protocol were used and the videos were coded by independent undergraduate or graduate research assistants. For all coded videos, treatment integrity was higher than 75%, with an average score of 90%.

## 3. Measures

Most of the questionnaires and observation methods we chose for our study were commonly used measures in PCIT evaluation research. Our primary outcome was the level of risk for child maltreatment, measured using the Brief Child Abuse Potential (BCAP) [39]. In all assessments, only mothers were involved. In addition to the standardized questionnaires, mothers completed a questionnaire for demographic information.

### 3.1. Brief Child Abuse Potential Inventory

The Child Abuse Potential Inventory (CAPI) [40,41] is a widely used and well-validated measure of the risk for child maltreatment. In the current study, we used the BCAP [39], a brief version including 34 items. Scores ranged from 0 to 34. For this version, similar psychometric properties were demonstrated. Parents indicated agreement or disagreement for the items and in our current study, we used the 24-item Child Abuse Risk Scale (BCAP Risk Scale), which is a total score of the subscales: distress, happiness, feelings of persecution, loneliness, family conflict, rigidity, and poverty. The subscales lie and random responding were not included in this total scale. The internal consistency (KR-20) in our sample of the Child Abuse Risk Scale was 0.93.

### 3.2. Eyberg Child Behavior Inventory

The Eyberg Child Behavior Inventory (ECBI) [42] is a 36-item parent rating scale that is well-validated and widely used to measure child disruptive behavior. The ECBI measures the frequency of child behavior problems (Intensity Scale) and the extent the parents experience found these behaviors problematic in children (Problem Scale). For the Intensity Scale, the response options range from one (*never*) to seven (*always*) and for the Problem Scale, a dichotomous scale (1 = *yes*, 0 = *no*) is used. For the Intensity Scale, scores range from 36 to 252 and for the Problem Scale, scores range from 0 to 36. The Dutch translation of the ECBI has well established reliability [43,44] and in the current study, internal consistencies were 0.94 for the Intensity Scale (Cronbach’s alpha) and 0.91 for the Problem Scale (KR-20).

### 3.3. Parenting Stress Questionnaire

To measure parenting stress, the Dutch Parenting Stress Questionnaire (Opvoedingsbelasting vragenlijst, OBVL) [45] was used. The 34-item OBVL is commonly used in Dutch outcome research and includes the following subscales: problems within the parent–child relationship, parenting problems, depressive moods, role restriction, and health problems. Total scale scores range from 34 to 136. In the present study, the *t* score of the sum of all items was used as the overall Parenting Stress Scale with an internal consistency (Cronbach’s alpha) of 0.70.

### 3.4. Dyadic Parent–Child Interaction Coding System

The Dyadic Parent–Child Interaction Coding System (DPICS) [46] is a behavior observation coding system that was used to measure the quality of parent–child interactions. In the current study, the DPICS was used to measure parenting skills during three 5-min structured situations: child-led play (CLP), parent-led play (PLP), and clean-up (CU). Each of these situations requires an increasing degree of parental control and direction. All DPICS observations were conducted with the mother and the child and for this study, the verbal behavior was observed and the frequencies were counted by independent coders. In our current study, two composite parent categories from the first structured situation (CLP) were used, which were derived for the comprehensive DPICS manual for research and training [46]. These categories included the percentage of positive following (coded in CLP only and including behavior descriptions, reflections, praises divided by the total of parent verbalizations), and percentage of negative leading (coded in CLP only including commands, questions, and negative talk divided by the total of parent verbalizations). Observations were coded by independent master-level research assistants and undergraduate students. All coders were intensively trained to 80% agreement with the first author. Additionally, all observations were transcribed and a minimum of one random situation per observation was coded again by a second coder to estimate interrater reliability. The overall percent agreement across the used DPICS categories was 85% (range 81–98%).

### 3.5. Therapy Attitude Inventory

To measure treatment satisfaction at post-treatment assessment, mothers completed the Therapy Attitude Inventory (TAI) [47]. This 10-item questionnaire measures the parental satisfaction of the process and outcomes of parent training, on a 5-point scale with higher scores indicating greater satisfaction. Questions focus on the parent’s perceptions about the discipline techniques learned, the quality of the parent–child interaction, change in the child’s behavior, and the overall family adjustment. Scores range from 5 to 50. Psychometric evaluation of the TAI demonstrated adequate reliability and validity [48] and the internal consistency (Cronbach’s alpha) of the TAI in the current study was 0.90.

### 3.6. Statistical Analyses

For the primary analyses of treatment outcomes, the child and parenting outcomes of families were compared for the IT group and WL group using analysis of covariance with baseline scores as covariates (ANCOVA). Additionally, to account for attrition, missing values were replaced according to the last observation carried forward (LOCF) method. LOCF was conducted for a small group of families who completed the baseline assessment, but failed to complete the subsequent assessments. Additionally, the effect sizes were presented using the eta-squared statistic.

To provide additional results with regard to the outcomes for both treatment groups on all assessment points, a per-protocol analysis was conducted using paired samples *t*-tests to investigate whether PCIT-Home led to significant improvements on the primary and secondary outcomes between baseline, post-test, and follow-up assessment. Effect sizes were calculated using Cohen’s *d.*

Prior to all analyses, the relevant assumptions were verified. Although most variables were normally distributed, the data of the BCAP risk scale variables did not meet this criteria. However, additional analyses using linear regression for residualizing out the covariates and a subsequent *t*-test on the residualized post-test scores, revealed similar results. This indicated that the non-homogeneity of variance did not affect the results for the BCAP risk scale. In addition, no multivariate outliers in our data were found.

## 4. Results

### 4.1. Sample Characteristics

Among the 20 families enrolled in the study, 16 families (9 IT and 7 WL) completed the T_1_ or T_01_, which was the post-treatment assessment for the IT group and the post-waitlist assessment for the WL group (see study flowchart, Figure 1). In total, 17 families (85%) fully completed the 8-session treatment and completed the post-treatment assessment (T_1_). Three families dropped out of treatment; one family did not start with the treatment because the mother decided to start treatment for other mental health problems first. The other two families only engaged in the first phase of treatment (four sessions) before dropout. In one case, another treatment was needed and the second family did not need treatment anymore since the child’s disruptive behavior had already dropped below the clinical range. Because only a few families discontinued PCIT-Home, the attrition rate in our study was therefore low (15%). Additionally, 14 families completed the follow-up assessment (T_2_) two months after treatment completion.

Demographic information of the total sample and randomization group is shown in Table 1. The children in the study were mostly boys (70%) with a mean age of 5.7 years old (*SD* = 1.6). Most children (60%) had a Dutch ethnicity; the other children had a non-Western background (Brazil, China, Ghana, Iran, Morocco, and Surinam). Given the large number of non-Dutch families, some parents were not native in Dutch or English. However, all parents, except one, mastered the Dutch or English language sufficiently to participate in treatment. One parent received treatment in French, which was possible since one of the therapists and a research assistant were fluent in French. The IT group included six foster care families compared to only biological families in the WL group. Based on the criteria of the Maltreatment Classification System (MCS) [49] for screening the child’s records at referral, 70% of the children had been exposed to any subtype of maltreatment at any time in their lives including physical abuse, sexual abuse, emotional maltreatment, physical neglect of basic needs, and physical neglect by lack of supervision. Since a number of children in this study lived in a foster family, not all children received treatment with the parent that was the suspected perpetrator. There were no significant differences in demographic variables between groups, except for the type of caregivers (two-tailed Fisher’s exact test, Table 1).

### 4.2. Immediate Treatment Compared to Waitlist

Mean scores on primary and secondary outcomes of the IT and WL groups, the results of the ANCOVA, and effect sizes are shown in Table 2. Prior to the primary analyses, the adequacy of the randomization was determined by comparing the baseline assessment scores on the outcomes between the IT and WL groups. Independent sample *t*-tests revealed significant differences between IT and WL groups at baseline assessment on parenting stress (OBVL, *t* (18) = −2.276, *p* = 0.035) and child abuse potential (BCAP risk scale, *t* (10) = −2.944, *p* = 0.015). Besides these variables, no significant differences were found on baseline assessment outcomes between the IT and WL families. Since the type of caregiver (biological vs. foster parent) significantly differed between groups, the analyses were conducted with this variable as a covariate.

Overall, from baseline assessment to treatment completion (eight sessions), the families who directly started with PCIT-Home (IT group) showed more improvements than the families who received no intervention for the initial two-month period (WL group). Specifically, large effect sizes (partial eta squared) were found for the decrease in child behavior problems (ECBI Intensity and Problem Scale). Additionally, even larger effect sizes were found in the IT group for the observed parenting skills. Parents who received PCIT-Home used more praise, behavior descriptions, and reflections (DPICS Positive Following) during play with their child and fewer questions, commands, and negative talk (DPICS Negative Leading) compared to parents in the WL group. However, no significant differences were found on the risk for child maltreatment (BCAP risk scale) and parenting stress (OBVL) between the IT and WL groups when the changes in this outcome were compared. Additional analyses including the LOCF for the small number of missing values did not change the significant outcomes.

### 4.3. Overall Treatment Effectiveness

In addition to the comparisons with the IT and WL groups, a per-protocol analysis was conducted for all assessment points. Table 3 shows the unadjusted means, effect sizes, and within group comparisons from the paired sample *t*-tests assessing the improvements over time for the total sample and for each treatment condition separately. For the total sample, compared to baseline scores, significant improvements after receiving PCIT-Home (T_1_) were found on all primary and secondary outcome measures, except for child abuse potential (BCAP risk scale), with moderate (0.54; OBVL-total) to high effect sizes (1.85; DPICS Negative Leading). Additionally, all these significant improvements were maintained at the follow-up assessment (T_2_), two months after termination of PCIT-Home. Additionally, a significant change for the total sample between baseline and follow-up was found for child abuse potential (BCAP risk scale, *t* (12) = 2.694, *p* = 0.020).

With regard to improvements within treatment conditions, outcomes were again generally positive with some exceptions. Significant reductions between baseline and post-test assessment on parenting stress, child abuse potential, and significantly less negative parental leading during the parent–child interaction were only found for families that directly started PCIT-Home after referral (IT group). Analyses were repeated to correct for the effect of differences between the groups in types of caregivers, but all outcomes remained unaffected.

### 4.4. Treatment Satisfaction

In order to measure parental satisfaction with PCIT-Home, mean scores on the TAI were calculated for the total sample. Since the TAI was only administered when families completed PCIT-Home, there were no data available from families who prematurely dropped out of treatment. The mean score of the 17 families that completed treatment was 41.47 (*SD* = 4.85), ranging from 32 to 50 (highest possible score), indicating an overall high treatment satisfaction. As expected, no differences between the IT and WL groups on TAI score were found, *t* (15) = −0.023, *p* = 0.982.

## 5. Discussion

The current study aimed to extend the knowledge about effective intervention for families that are at risk for child maltreatment as well as explore ways of adapting standard treatments for use in the community and increase their accessibility to improve public health. With this study, our goal was to evaluate the effectiveness of PCIT-Home, a home-based and time-limited version of PCIT, by conducting a randomized controlled trial using a waitlist control condition in an at-risk family population who were referred to a Dutch community mental health center. By providing a brief home-based version of PCIT, we expected a lower attrition rate compared to research studies evaluating standard PCIT. Additionally, we expected reduced child abuse potential, disruptive behavior, and parenting stress as well as increased parenting skills. Subsequently, we expected these factors to prevent child maltreatment.

In line with the expectations, the results of the IT group compared to the WL group showed positive effects on the reduction of child disruptive behavior and the increase in parental skills. Since PCIT focuses directly on parenting skills, which showed the largest effect sizes, those increased skills likely modified the disruptive behavior of the child. These results suggest the added value of PCIT-Home compared to the WL group; no (immediate) treatment. Additionally, the positive effects of PCIT-Home for the total sample remained at the two-months follow-up. However, no significant differences were found in parenting stress and the risk for child abuse between the IT and WL groups. A possible explanation for these findings is that a decrease in parenting stress and the risk for child maltreatment are secondary effects that follow from changes in parenting skills and child behavior and could take more time to surface than the total four-month assessment period in our study. In addition, our measures for parenting stress (OBVL) and child abuse potential (BCAP) both included a variety of characteristics such as depressive moods, health problems, family conflicts, and poverty that are not likely targeted in this brief version of PCIT. Given this consideration, our assessment period and the sample size, it might not be realistic to expect short-term effects on all outcome measures. This could be an explanation for why we did not find significant differences in parenting stress and child abuse potential measures variables. However, when the scores of all the families who received PCIT-Home were compared at baseline and follow-up assessments, we did find a treatment effect on the reduction of child abuse potential.

Comparing groups when they both received PCIT-Home, our results showed that the IT group seems to have more positive effects of the intervention, specifically on parenting stress and the risk for child maltreatment. A possible explanation is that parents find more benefit in the immediate start, because in the period after referral, they may need help the most. Another possible explanation could be a consequence of the initial differences between the treatment groups, in which foster parents were overrepresented in the IT group. The findings hint at different types of challenges that biological and foster parents may experience in dealing with the child’s behavior problems and in the process of being coached during PCIT. Furthermore, biological and foster parents may have different histories with the children and perhaps different types of personal characteristics that may influence their ability to benefit from the program. Thereby, in clinical practice, therapists typically pay attention to these individual differences among parents and can take those into account during the sessions, even when delivering a standardized treatment such as PCIT.

Additionally, as expected, the attrition rate found in this study (15%) was low compared to other PCIT research studies [21,50], indicating higher accessibility of the intervention. In addition, this percentage was substantially lower than the 40% attrition rate found in a previous Dutch study evaluating clinic-based PCIT [22]. Since the results on lower rates of treatment attrition have been inconclusive based on previous studies, the combination of a brief home-based PCIT version might have contributed to this low attrition rate in our study. Thereby, the families who prematurely dropped out in our study did not discontinue treatment because of specific barriers to treatment such as low motivation or practical issues.

### 5.1. Clinical Implications

Providing treatment at the family’s home creates a higher accessibility of interventions for families. In particular, families facing multiple problems and who are at high risk for dropout could benefit from a home-based treatment. Additionally, the time-limited eight sessions in PCIT-Home, with a clearly defined end-point probably encouraged families to stay engaged in treatment. Regarding the improved accessibility by a low attrition rate, our study showed high effectiveness of PCIT-Home.

However, some considerations have to be taken into account for the use of PCIT-Home in clinical practice. Providing PCIT-Home led to a higher time investment of therapists per treatment session. Because therapists need to travel to the family, they have less time to see other clients. Since the financial system in the Dutch community mental health practice is generally based on direct time with a client, this can lead to negative consequences for the productive time of therapists. This could be a financial barrier for agencies to initially implement and provide a home-based treatment. Nevertheless, the higher accessibility of home-based intervention also implies higher cost-effectiveness, emphasizing the need to also evaluate this in future research.

Despite a small number, some parents rated their child’s behavior still within a clinical range after PCIT-Home, suggesting the need for more sessions. A clinic playroom without the many stimuli and distractions from the home situation could help parents to learn skills faster, with more effect on the child’s behavior problems. In addition, in the case of serious disruptive behavior of the child, a playroom in the clinic with higher environmental control might be a safer option to practice time-out. Further use and implementation of PCIT-Home would therefore benefit from additional screening at referral, for example, to only provide PCIT-Home for families at high risk for dropout by assessing their motivation at the intake session and their practical barriers to come for treatment in the clinic. Additionally, in the case of serious child behavior problems, therapists can easily move to the standard PCIT with no time limit since the treatment protocol of the PCIT-Home session does not differ from the standard PCIT.

### 5.2. Strengths and Limitations

The current study focused on an outcome that has been limited studied previously, namely the risk for child maltreatment. By evaluating an adapted version of an evidence-based treatment, our findings contribute to the accumulating knowledge to prevent child maltreatment. Our study also has high clinical representativeness [51], since it was conducted in a real-world clinical context, with community referrals as participants and practicing clinicians as therapists. However, conducting research in a real-world clinical context often comes with limitations, and therefore some of them need to be mentioned. First, the primary limitation is the small sample size, which may have affected our outcomes. Thereby, despite randomization, the treatment groups had initial group differences on higher child abuse potential and higher parenting stress for the WL group at baseline assessment. These differences may have affected the impact of PCIT when testing for changes in the other outcome variables such as parenting skills. In addition, the IT group not only started with lower levels on child abuse potential, but their scores were also very low with low variance compared to the WL group, and that they had little room for improvement.

Although the purpose was to include at-risk families, the actual sample in our study was less at-risk than expected at the time of referral. Since a number of children lived in a foster family and did not receive treatment with the parent that was the suspected perpetrator of the child maltreatment; for these children, the risk for maltreatment was no longer present. Second, the various family situations (i.e., biological parents versus foster parents) may have influenced the generalizability of the current study. Despite the limited number of participants, significant differences were detected between treatment groups, indicating the added value of PCIT-Home in comparison with no treatment. In addition, compared to the existing literature, the current sample size is not uncommon for community-based PCIT evaluation studies [36].

A third limitation of the current study is related to the difficulties measuring the risk for child maltreatment. Therefore, our study could have benefited from using child abuse potential as the only inclusion criterion to select families with a high risk for child maltreatment and avoid low levels on this primary outcome at baseline assessment. In addition, the validity of the outcome measure (BCAP) is limited. In our study, the assumption of normality was violated for this variable, which may have affected the statistics. Furthermore, standardization of the BCAP for the Dutch situation is lacking and other measures appeared to be not suitable to measure the effects of PCIT on the risk for child maltreatment in a reliable way. This corresponds with the existing literature about the validity of measures to predict the risk for child maltreatment [52]. Based on this limitation, drawing conclusions on the prevention of the risk for child maltreatment is therefore limited. However, the differences found on outcomes associated with lower risk for child maltreatment such as decreased child disruptive behavior problems and improved parenting skills indicate the potential of PCIT-Home to be an effective treatment to prevent child maltreatment.

The last limitation worth mentioning is that the period between the post-treatment assessment and the follow-up assessment was short. In other words, the long-term effects of PCIT-Home have not been examined yet and additional research is needed to investigate the long-term gains of this brief intervention. Based on the limitations, additional research addressing these limitations is recommended.

## 6. Conclusions

Despite the limitations, our current study shows promising findings in the field of the prevention of child maltreatment by creating higher accessibility of treatment through providing a brief and home-based version of PCIT for at-risk families who may not have access otherwise. The study does show some encouraging evidence that PCIT can be implemented effectively in a real-world community setting, with apparently low attrition. However, additional evaluation research is recommended to support our findings that reduced child disruptive behavior problems and improved parenting skills also effectively reduce the risk for child maltreatment. Although our study did only find indirect evidence for the prevention of child maltreatment, our findings did contribute to the knowledge on evidence-based practice in the prevention of child maltreatment.

## Figures and Tables

**Figure 1 ijerph-18-08244-f001:**
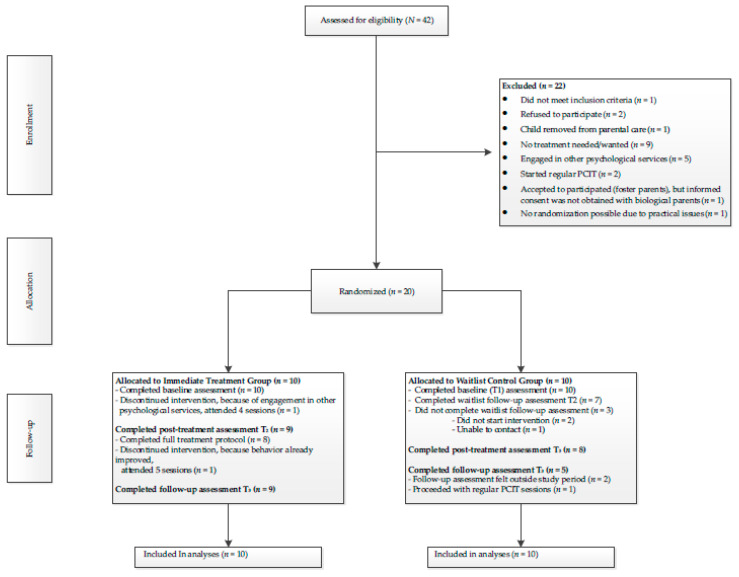
Study flowchart.

**Table 1 ijerph-18-08244-t001:** Demographic information for the total sample and by randomization group.

	Means (SD) or Percentages			
	Total (*N* = 20)	IT (*n* = 10)	WL (*n* = 10)	*p*
**Child characteristics**				
Age (years)	5.7 (1.6)	5.8 (1.9)	5.6 (1.3)	0.795
Gender (% male)	70.0	60.0	80.0	0.628
Ethnicity (% non-Western)	40.0	40.0	40.0	1.00
Maltreatment history (% reported in client file)	70.0	90.0	50.0	0.141
**Family characteristics**				
Caregivers (% biological parent)	70.0	40.0	100.0	0.011 *
Family status (% single-parent)	40.0	30.0	50.0	0.650
Family income (% poor or fair financial situation)	45.0	20.0	70.0	0.070
**Therapy characteristics**				
Dropout (% that not completed the 8-session protocol)	15.0	10.0	20.0	1.00

*p* = probability of differences between samples according to independent samples *t*-tests and chi-square test (or Fisher’s exact tests). IT: immediate treatment group; WL: waitlist control group; ** p* < 0.05.

**Table 2 ijerph-18-08244-t002:** Comparisons of change between outcomes at time 1 and time 2.

			Baseline (T_0_)			Post-Test (T_1_/T_01_)				
Measures	Group	*n*	*M*	*SD*	*n*	*M*	*SD*	*F (1,10–14)*	*p*	$\eta_{p}^{2}\;$
**Child behavior**										
ECBI Intensity	IT	10	121.0	30.2	9	80.8	24.9	13.12	0.003 *	0.502
	WL	10	131.3	40.6	8	136.3	46.0			
ECBI Problem	IT	10	13.6	8.3	9	5.6	4.4	5.44	0.038 *	0.312
	WL	9	19.4	7.5	7	19.4	10.4			
**Child abuse potential**										
BCAP risk scale	IT	10	2.2	1.7	9	1.0	1.3	0.78	0.392	0.057
	WL	10	9.1	7.2	8	7.5	6.2			
**Parenting stress**										
OBVL-total *(t-score)*	IT	10	59.9	11.7	9	48.7	8.3	2.46	0.141	0.159
	WL	10	69.6	6.6	8	67.6	9.3			
**Parenting skills**										
DPICS % Positive Following	IT	9	9.7	7.9	8	36.0	12.1	18.30	<0.001 *	0.604
	WL	10	4.7	4.2	8	6.5	5.4			
DPICS % Negative Leading	IT	9	38.0	12.9	8	11.4	10.3	15.55	<0.002 *	0.564
	WL	10	40.9	15.0	8	39.3	10.7			

Post-test assessment is T_1_ for the IT group and T_01_ for the WL group. ** p* < 0.05.

**Table 3 ijerph-18-08244-t003:** Unadjusted means, effect sizes, and within- and between group comparisons for the total sample.

			Pre-Test (T_0_/T_01_)		Post-Test (T_1_)				Follow-Up (T_2_)			Effect Size *d*
Measures	Group	*n*	*M*	*SD*	*M*	*SD*	*p (T_0_–T_1_)*	*n*	*M*	*SD*	*p (T_1_–T_2_)*	Within-Group T_0_–T_2_
**Child behavior**												
ECBI Intensity	IT	9	116.2	27.7	80.8	24.9	0.001 *	9	84.1	28.0	0.206	1.15 *
	WL	8	138.4	43.4	103.3	46.2	0.040 *	5	101.2	28.6	0.698	1.01 *
	Total	17	126.6	36.6	91.4	37.1	<0.001 *	14	90.2	28.4	0.310	1.65 *
ECBI Problem	IT	9	11.9	6.7	5.6	4.4	0.030 *	9	3.9	3.1	0.236	1.53 *
	WL	8	18.6	9.9	10.8	6.6	0.029 *	5	12.8	5.1	0.675	0.74 *
	Total	17	15.1	8.8	8.0	6.0	0.001 *	14	7.1	5.8	0.396	1.07 *
**Child abuse potential**												
BCAP risk scale	IT	9	2.1	1.8	1.0	1.3	0.030 *	8	1.3	1.8	0.763	−0.44 *
	WL	8	7.3	6.3	7.1	6.1	0.912	5	7.6	6.8	0.178	−0.05
	Total	17	4.5	5.1	3.9	5.2	0.261	13	3.7	5.3	0.387	0.15 *
**Parenting stress**												
OBVL-total	IT	9	58.3	11.2	48.7	8.3	0.006 *	8	49.1	12.3	0.964	0.78 *
	WL	8	67.1	9.3	65.1	12.0	0.629	5	65.8	13.3	0.553	0.11
	Total	17	62.5	11.1	56.4	13.0	0.025 *	13	55.5	14.8	0.926	0.54 *
**Parenting skills**												
DPICS % Positive Following	IT	8	8.3	7.3	36.0	12.1	<0.001 *	8	31.3	15.6	0.269	−1.89 *
	WL	7	5.5	4.0	25.0	22.4	0.052 *	4	26.5	27.0	0.340	−1.09
	Total	15	7.0	5.9	30.9	17.9	<0.001 *	12	29.7	18.9	0.690	−1.62 *
DPICS % Negative Leading	IT	8	38.5	13.6	11.4	10.3	<0.001 *	8	17.0	11.8	0.035*	1.69 *
	WL	7	39.3	10.7	23.2	16.7	0.132	4	15.1	11.0	0.419	2.23 *
	Total	15	37.7	12.0	16.9	14.5	<0.001 *	12	16.3	11.1	0.987	1.85 *

For DPICS % Positive Following high means indicate improvement. Asterisks in the T_0_–T_2_ within-group effect size column indicate significant change from baseline to follow-up. * *p* < 0.05.

## Data Availability

Data available on request due to restrictions (e.g., privacy or ethical). The data presented in this study are available on request from the corresponding author. The data are not publicly available due to privacy and ethical guidelines.

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
