# Peer review of "Home-Based Parent–Child Interaction Therapy to Prevent Child Maltreatment: A Randomized Controlled Trial"

_ijerph, 2021, doi:10.3390/ijerph18168244_

Round 1

Reviewer 1 Report

I appreciate this contribution to community based intervention research addressing an important clinical and community concern.  Overall the authors have presented this work well;  the detail of measures and thorough accounting of limitations were especially well done.  A few english language corrections are suggested: Pg 1 ln 38 "leaded" to "led"; Pg 4 ln 160 "simply" to "simplify"; Pg 6 ln 203 "strategical" to "strategic"; Pg 6 ln 207 "was is" choose one; Pg 6 ln 233 "scores ranges" to "scores range".

Author Response

Thank you for your thorough reading of the manuscript. We did modify the suggested English language corrections and also read the manuscript carefully again and made additional language corrections to improve the readability of the manuscript.

Reviewer 2 Report

Commentary

Method

The reliability of the test should be calculated by factors. The Cronbach's Alpha value is vulnerable to the length of the test.

Cronbach's Alpha assumes normality of the data, how the authors justify using the statistic without verifying this assumption.

Results

The authors use parametric tests but do not indicate how they verified the data for normality.

To compare groups 1 and 2, they used an ANCOVA test. I do not think it is appropriate to use it in this case. I think it is suitable in this case. I get the impression that a Student's t-test would be convenient. The effect size in the analyzes of variance is usually determined with the eta-squared statistic.

Results

The authors use parametric tests but do not indicate how they verified the data for normality.

To compare groups 1 and 2, they used an ANCOVA test. I do not think it is appropriate to use it in this case. I think it is suitable in this case. I get the impression that a Student's t-test would be convenient. The effect size in the analyzes of variance is usually determined with the eta-squared statistic.

Author Response

Thank you for your thorough reading of our manuscript and the suggestions for adaptation for our statistical methods. Below we explain how we considered these comments:

Method

  1. The reliability of the test should be calculated by factors. The Cronbach's Alpha value is vulnerable to the length of the test.

Thank you for this suggestion. The questionnaires we have chosen for our study were commonly used measures in PCIT evaluation research. We understand that calculating reliability / internal consistency of the test by factors is favorable. However, in order to compare the values of internal consistency with other studies we prefer to keep using Cronbach’s Alpha since these values are used in other treatment effectiveness studies on PCIT. Also, for all measures the Cronbach’s Alpha (or KS-20) is used to report internal consistency in the psychometric evaluation studies.

  1. Cronbach's Alpha assumes normality of the data, how the authors justify using the statistic without verifying this assumption.

Thank you for this comment. We agree that verifying the assumption of normality of the data distribution is important. Prior to all statistical analyses we verified the associated assumptions of the tests. Since we did not make it clear in our manuscript, we did verify the assumptions we included this to the manuscript.

Results

  1. The authors use parametric tests but do not indicate how they verified the data for normality.

As described in the answer to the previous comment, we did verify the assumptions and we also conducted additional analyses for variables that were not normal distributed. However, this was not mentioned in the manuscript. Therefore we added an additional paragraph to the Statistical Analyses section, (Pg 8 ln 299):

“Prior to all analyses the relevant assumptions were verified. Although most variables were normal distributed, the data of the BCAP risk scale variables did not meet this criteria. However, additional analyses, using linear regression for residualizing out the co-variates and a subsequent t-test on the residualized post-test scores, revealed similar results. This indicated that the non-homogeneity of variance did not affect the results for the BCAP risk scale.”

  1. To compare groups 1 and 2, they used an ANCOVA test. I do not think it is appropriate to use it in this case. I think it is suitable in this case. I get the impression that a Student's t-test would be convenient. The effect size in the analyzes of variance is usually determined with the eta-squared statistic.

Thank you for this suggestion. Although we are not completely sure, we think you are addressing our primary analysis for the comparison between the IT group and WL group. To choose the appropriate test for our study and data, we have consulted an statistical expert. He did recommend using an ANCOVA test for our primary analysis, because of the possibility to control for the baseline scores and differences at baseline between groups. We also conducted a Student’s t-test which showed in general no different outcome. For this reason, we decided to keep the ANCOVA test. We additionally thank you for the suggestion to use another statistic (eta-squared) for the effect size. We agree that this statistic is more appropriate for the analyses of variance. We therefore replaced the calculated Cohen’s d in Table 2 into partial eta squared values. The interpretation of the effect sizes remained the same.

Round 2

Reviewer 2 Report

Could be reported reliability by factor and general. It is not clear to me why the authors did not attend the suggestion.

Is not enough to say "verified the assumptions. The authors can be explained how they examined multivariate normality?

When the scales are multidimensional, the reliability of each dimension must be reported. You can review these articles.

See description of the empathy toward victims scale (two factors cognitive and affective empathy). Article: Vlaanderen, A., Bevelander, K. E., & Kleemans, M. (2020). Empowering digital citizenship: An anti-cyberbullying intervention to increase children´s intentions to intervene on behalf of the victim. Computer in Human Behavior, 112, Article e106459. doi:10.1016/j.chb.2020.106459   Collective Efficacy Scale (two subscales Afterschool connectedness subscale and willingness to intervene subscale). Article: Smith, E. P., Osgood, D. W., Caldwell, L., Hynes, K., & Perkins, D. F. (2013). Measuring collective efficacy among children in community-based afterschool programs: Exploring pathways toward prevention and positive youth development. American Journal of Community Psychology, 52, 27-40. doi: 10.1007/s10464-013-9574-6   The authors acknowledge that the BCAP variable did not meet the normality criteria.  The authors should consider that the violation of the assumption of normality can affect the estimates of the statistics used in the study.   However, I consider that the article can already be published

Author Response

Please find the attached file with our point-by-point response to your comments.
